# Preparation and Characterization of Antioxidative and pH-Sensitive Films Based on κ-Carrageenan/Carboxymethyl Cellulose Blended with Purple Cabbage Anthocyanin for Monitoring Hairtail Freshness

**DOI:** 10.3390/foods14040694

**Published:** 2025-02-18

**Authors:** Manni Ren, Ning Wang, Yueyi Lu, Cuntang Wang

**Affiliations:** 1College of Food and Bioengineering, Qiqihar University, Qiqihar 161006, China; 2Engineering Research Center of Plant Food Processing Technology, Ministry of Education, Qiqihar 161006, China

**Keywords:** pH-sensitive indicator films, purple cabbage anthocyanin, carrageenan, carboxymethyl cellulose, hairtail freshness

## Abstract

Developing pH-sensitive materials for real-time freshness monitoring is critical for ensuring seafood safety. In this study, pH-responsive indicator films were prepared by incorporating purple cabbage anthocyanin (PCA) into a κ-carrageenan/carboxymethyl cellulose (CA/CMC) matrix via solution casting, with PCA concentrations of 2.5%, 5.0%, 7.5%, and 10% (*w/w*). The films exhibited remarkable pH sensitivity, with distinct color changes across pH 2.0–11.0. Incorporating PCA enhanced film crystallinity, antioxidant properties, and opacity while reducing water vapor transmission (WVP). High PCA content resulted in rougher morphology, lowering tensile strength (TS) but improving elongation at break (EB). The indicator film had good environmental stability, and the color difference was not visible after 10 days in the dark and 4 °C conditions. The CA/CMC/PCA-10% film showed the most pronounced pH-responsive color changes, transitioning from purple to green as hairtail freshness deteriorated. This innovative approach highlights the potential of CA/CMC/PCA films as reliable, eco-friendly indicators for real-time seafood freshness monitoring, offering significant advancements in smart packaging technology.

## 1. Introduction

The global food industry is tackling a critical challenge: reducing food and packaging waste without compromising food quality and safety [1]. Among various food types, seafood products are particularly demanding regarding adequate packaging due to their perishable nature, which often results in significant storage and transportation challenges [2]. During storage, seafood products are susceptible to bacterial contamination, producing spoilage substances, such as histamines, which affect food safety and pose a risk of foodborne illness. Furthermore, a notable feature of seafood spoilage is the change in pH, as the seafood flesh shifts from acidic to alkaline during decay, resulting in a rise in pH value [3]. As such, developing smart packaging materials that can monitor pH changes in real time and provide feedback on food quality has become a key research focus for ensuring the freshness and safety of seafood products. As indicator packaging technology, pH-sensitive smart packaging can display color changes to signal pH fluctuation, thus offering consumers and producers intuitive feedback on the spoilage status of the food.

pH-responsive indicators can be classified into synthetic chemical indicators and natural pigment indicators. While synthetic chemical pH indicators offer advantages such as rapid response, significant color change, and good stability, they may pose potential safety risks due to migration into food [4]. In recent years, natural pH-responsive indicators, such as curcumin, anthocyanins, alizarin, and carotenoids, have attracted increasing attention due to their natural origin, safety, wide availability, and prominent color-changing properties [5]. Natural pigments have been widely used in food packaging as pH-responsive indicators to monitor seafood freshness [6,7,8,9].

Anthocyanins are a group of water-soluble flavonoids commonly found in fruits and vegetables like blueberries and purple cabbage. They exhibit significant pH-responsive properties and antioxidant activity [10]. These compounds undergo noticeable color changes under different pH conditions, making them ideal pH indicators. Under acidic conditions, anthocyanins generally appear red; in alkaline conditions, they shift to blue or green. This color change is primarily caused by the binding and dissociation of hydrogen ions (H⁺) in the anthocyanin molecule, leading to structural changes [11]. Compared to lipid-soluble indicators, such as curcumin, anthocyanins are more versatile due to their water solubility, making them easier to use in food packaging applications. Purple cabbage (*Brassica oleracea* L.) is a rich source of anthocyanins. It is low cost, high yield, and contains abundant anthocyanin content. Purple cabbage anthocyanin (PCA) is a natural, water-soluble pigment extracted from purple cabbage. Its molecular structure consists of an aromatic ring with multiple phenolic hydroxyl groups connected to sugar molecules via glycosidic bonds to form anthocyanins [12]. The significant color change of purple cabbage anthocyanins under varying pH conditions and their antioxidant, anti-inflammatory, and anti-tumor activities make them highly promising for food freshness monitoring and smart packaging applications. For instance, Kan et al. (2022) extracted anthocyanins from 14 different plant sources and incorporated them into starch/polyvinyl alcohol matrices to develop a series of smart packaging labels for monitoring the freshness of pork and shrimp [13]. The findings revealed significant variations in anthocyanin and phenolic content among plant sources, notably influencing the packaging labels’ performance. Labels incorporating PCA were particularly effective for pork freshness monitoring, while those containing extracts from *Lyceum ruthenicum* and black eggplant peels were more suitable for shrimp freshness monitoring. Pang et al. (2023) prepared a gelatin film incorporated with PCA to monitor the freshness of fish [7]. The results indicated that the addition of PCA could endow the gelatin film with good UV resistance and antioxidant properties. The film containing 0.28% of PCA could extend the shelf life of chilled-stored fish.

Petroleum-based plastic films and trays are mainly used for packaging seafood products. These materials have excellent mechanical properties and stability. However, they are non-renewable and non-biodegradable, causing severe environmental pollution. In this context, natural polymer materials have become selective due to their biodegradability and biocompatibility. Polysaccharides such as starch, cellulose, chitosan, and κ-carrageenan (CA) are widely used in food packaging [1]. CA is a sulfated polysaccharide extracted and isolated from red algae with good film-forming ability. However, the film prepared from pure CA has certain defects in mechanical properties [14]. Carboxymethyl cellulose (CMC) is a derivative of cellulose. Blending it with CA can improve its mechanical properties. Hamdan et al. (2021) and Kang and Yun (2023) prepared CA/CMC composite films or hydrogels and found that an interpenetrating network formed between anionic CMC and CA molecules, thus improving the mechanical properties of the composite materials [15,16].

This study aims to develop a smart bio-based packaging film with real-time spoilage monitoring capabilities by incorporating PCA into CA/CMC composite films. It is hypothesized that PCA will significantly enhance the pH sensitivity and antioxidation of the composite films, enabling them to track spoilage in hairtail fish effectively. The effects of PCA incorporation on the morphology, structure, optical, mechanical, antioxidant, and pH-dependent color-changing properties of the composite films were investigated. Furthermore, the potential practical application of composite films in food packaging was also evaluated. This study provides novel insights into the functional development of smart packaging films with real-time indicators and contributes to enhancing the safety and intelligence of food packaging.

## 2. Materials and Methods

### 2.1. Materials and Reagents

Fresh purple cabbage (*Brassica oleracea* L.) and hairtail fish (*Trichiurus lepturus*) were purchased in Darunfa supermarket. κ-Carrageenan (food-grade) was purchased from Qingdao Dehui Marine Biotechnology Co., Ltd. Carboxymethyl-cellulose (CMC) (MW = 262.19000, DS = 0.92) was purchased from Hebei Qianju Biotechnology Co., Ltd. DPPH radical and ABTS radical were purchased from Sigma Chemical Co. (Saint Louis, MO, USA). All other reagents were of analytical grade.

### 2.2. Extraction of Anthocyanins from Purple Cabbage

The extraction of anthocyanin from purple cabbage was carried out with slight modifications referring to the method of Jiang et al. (2020) [17]. The purple cabbage anthocyanin (PCA) was reserved in −20 °C for use. The total anthocyanin content in the obtained powder was (92.5 ± 4.2) mg g^−1^, determined by a pH-differential assay (Zhang et al., 2020) [18].

### 2.3. Preparation of Films for CA/CMC/RA

The CA/CMC/PCA pH-sensitive indicator films were prepared by solution casting according to the method of Wang et al. (2024) [19]. The films were designated as CA/CMC, CA/CMC/PCA-2.5%, CA/CMC/PCA-5%, CA/CMC/PCA-7.5%, and CA/CMC/PCA-10% based on the incorporation levels of PCA.

### 2.4. pH Response and UV-VIS Spectroscopic Determination of PCA

The determination of the UV-visible spectrum of PCA was carried out by referring to the method of Zhang et al. (2020) with slight modifications [18]. Briefly, PCA (10 mg) was dissolved for 20 min in 10 mL deionized water. The pH of the PCA solution was adjusted to the range of 2.0 to 11.0 using 1 mol/L hydrochloric acid or 1 mol/L sodium hydroxide. Using an ultraviolet-visible spectrophotometer, the scanning wavelength range was set from 400 to 800 nm, with distilled water as the blank control.

### 2.5. Determination of Fourier Transform Infrared (FT-IR) Spectroscopy

FT-IR spectroscopy of CA/CMC/PCA films was measured according to the method of Gao et al. (2021) [20].

### 2.6. X-Ray Diffraction Analysis

XRD analysis of CA/CMC/PCA films was measured according to the method of Wang et al. (2022) [21].

### 2.7. Scanning Electron Microscope (SEM)

The microstructure of CA/CMC/PCA films was analyzed by SEM (Model S-4300, Hitachi Corporation, Tokoy, Japan) according to the method of Gao et al. (2021) [20].

### 2.8. Color Parameters and Opacity

The color parameters of the CA/CMC/PCA films were established by adhering to the methodology proposed by Gao et al. (2021) [20]. The L*, a*, and b* of the CA/CMC/PCA films were quantified using a colorimeter (LS173, Shenzhen LinSheng Technology Co., Ltd.).

The optical properties of the CA/CMC/PCA films were ascertained in accordance with the method of Sukhija et al. (2016) [22].

### 2.9. Determination of Mechanical Properties

The thickness of the CA/CMC/PCA films was determined according to the method of Liu et al. (2021) [23]. Mechanical properties of CA/CMC/PCA films were determined according to the method by Mu et al. (2012) [24].

### 2.10. Determination of Water Vapor Permeability (WVP)

WVP of CA/CMC/PCA films were determined following to the method of Crizel et al. (2017) [25].

### 2.11. Determination of Total Phenolic Content and Radical Scavenging Properties

The total phenolic content of the CA/CMC/PCA films was determined by Gao et al. (2021), and the results were expressed as GAE mg/dw/g (GAE: gallic acid) [20]. The radical (ABTS and DPPH) scavenging capability of CA/CMC/PCA films was determined by the method of Gao et al. (2021) [20]. Free radical scavenging rates (%) were calculated.

### 2.12. Indicates the pH Sensitivity of the Film

The pH sensitivity determination of the CA/CMC/PCA films was determined by the method of Choi et al. (2017) with minor adjustment [26]. At 25 °C, the indicator films cut into 2 cm × 4 cm were respectively immersed in pH buffer solutions ranging for 2 to 11 for 10 min, and the colors of the films were photographed.

### 2.13. Determination of Color Stability

The determination of color stability of the CA/CMC/PCA indicator film refers to the method of Bao et al. (2022) [27].

### 2.14. Application of CA/CMC/RA in Spoilage Monitoring of Hairtail

Color change of the indicator film during hairtail storage was determined by the method of Wang et al. (2024) [19]. Twelve frozen hairtail meat samples were mixed. Hairtail meat samples (10.00 g) were placed in a petri dish (diameter: 90 mm; height: 15 mm). The 2 × 4 cm composite film was adhered to the lid of the plate and the plate was sealed with plastic wrap. This sealed petri dish was stored at 4 °C. The TVB-N and pH were detected regularly (every 48 h), and colors of indicator films were photographed simultaneously. TVB-N and pH of the hairtail fish samples were determined by the method of Ge et al. (2020) [28].

### 2.15. Statistical Analysis

All experiments were carried out three times, and the results were expressed using the mean ± standard deviation. The experimental data were analyzed for significant differences using Duncan’s in the SPSS 26 system software, and the drawing was carried out using the Origin (version 8.5) software.

## 3. Results and Analysis

### 3.1. pH Response and UV-VIS Spectrum of PCA

In ultraviolet-visible spectroscopy analysis, anthocyanins exhibit significant absorbance changes [29]. The absorbance changes at different wavelengths can reflect the different structures and properties of anthocyanins. Anthocyanins from various plant sources have different sensitivities to pH value changes [29]. In different pH solutions, anthocyanins display different chemical structures and colors [22]. Thus, the pH sensitivity of anthocyanins is a crucial factor in the preparation of indicator films. The sensitivity of PCA’s color response in solutions of different pH values is closely related to its concentration. In aqueous solutions, a low concentration of PCA (1 mg/mL) can already exhibit significant color changes. Since the color of PCA extract is deep purple, high concentrations of PCA solution can mask the color changes under different pH conditions. The color and ultraviolet-visible spectroscopy diagram of anthocyanins from purple cabbage at different pH values is shown in Figure 1. When the pH was less than 3, the PCA solution was pink. When the pH ranged from 4.0 to 5.0, it gradually turned purple. As the pH increased from 6.0 to 9.0, the color changed from blue to green. When the pH was between 10.0 and 11.0, it turned yellow. The color change of anthocyanins at different pH values was caused by their structural differences. This is because changes in acidity and alkalinity disrupt the balance among the four structures in anthocyanins: flavylium cation, quinone base, carbinol pseudobase, and chalcone. In a strongly acidic environment, the flavylium cation is dominant, so the solution appears red due to the flavylium cation. As alkalinity rises, the flavylium cation gradually changes into the colorless carbinol pseudobase, and the red color of the solution weakens accordingly. As alkalinity keeps increasing, the quinone base gradually becomes the main structure, making the solution change color. Finally, because the environment is too alkaline, anthocyanins are unstable and degrade, and the solution turns yellow [30]. Ultraviolet-visible spectroscopy was used to characterize the color changes of PCA in different pH buffer solutions. In the ultraviolet-visible spectroscopic analysis, anthocyanins showed obvious absorbance changes. The absorbance changes at different wavelengths can reflect the different structures and properties of anthocyanins. Figure 1 shows the ultraviolet-visible spectra of PCA in various pH buffer solutions. As the pH value increased from 2.0 to 11.0, the maximum absorption peak gradually shifted from OD 537 nm to 561 nm. At the same time, the maximum absorption intensity of PCA gradually decreased as the pH decreased. In acidic conditions, a relatively high peak was observed but as the pH value increased to 6.0, its height gradually decreased. When the pH value reached 9.0, the intensity increased and reached a new peak. These significant color changes suggest that PCA can be used for the development of indicator films. Research on blueberry anthocyanin [27] and blackcurrant anthocyanin [30] indicator films also found that anthocyanins show strong color variations in different pH buffer solutions.

### 3.2. Fourier Infrared Spectrum

FT-IR spectroscopy was used to study the composite films. The molecular interactions between anthocyanins and the film matrix were determined by the specific frequencies at which different chemical bonds or functional groups absorbed infrared light [31]. As seen in Figure 2, all the films had similar band patterns. The CA/CMC composite film had characteristic spectral bands at 3300 cm^−1^ (O-H stretching) and 2924 cm^−1^ (C-H stretching). The peak at 1592 cm^−1^ was from the stretching vibration of C-O. The deformation vibration of the C-H or N-H bond appeared at 1416 cm^−1^, and the absorption peak at 1148 cm^−1^ was due to the stretching vibration of the C-O bond and the bending vibration of the OH bond. There were three characteristic peaks at 1017, 920, and 845 cm^−1^. These were the stretching vibrations of the C-O-C bond and O-C bond in carrageenan, CMC, and the plasticizer respectively. The characteristic peaks of PCA were at 1416 cm^−1^ and 1017 cm^−1^, which were attributed to the C-O angular deformation of phenolic compounds and the C-O-C stretching vibration of the glucose dehydration ring respectively. After adding PCA to the film, a slight band shift was observed. This indicated the formation of hydrogen bonds between PCA and the CA/CMC composite. These results show that FTIR can be used to analyze the interactions between the components. Some researchers have suggested that the spectral band changes in the film with anthocyanins were related to the interactions between the film components. Zhang et al. (2020) found that when plant anthocyanins were added to the starch/PVA film, similar changes in band location and peak intensity were seen [18].

### 3.3. X-Ray Diffraction (XRD)

XRD can be used to characterize the crystal structure of the composite film. Crystallization performance is an important indicator that affects the properties of the complex film. It is closely related to the mechanical properties of the complex film [20]. When X-rays of a certain wavelength are irradiated onto crystalline substances, the X-rays scatter when they encounter the atoms or ions inside the crystal. Thus, a unique diffraction phenomenon corresponding to the crystal structure is presented [32].

By studying these diffraction patterns, we can learn about the makeup, structure, and shape of atoms or molecules inside the film materials [20]. Figure 3 shows how different amounts of PCA affect the crystal structure and compatibility of the complex films. A wide peak at 2θ = 21.00° is the special peak for carboxymethyl cellulose. This peak shows its unique crystal structure, which has low crystallinity and also shows that PCA is amorphous. The κ-carrageenan-mulberry polyphenol extract film has a similar amorphous structure [33]. Both the CA/CMC complex films and the CA/CMC/PCA films have the same diffraction peaks. They both have a strong peak at 2θ = 28.38° and weaker peaks at 9.34°, 28.38°, and 40.62°. This means the PCA extract spreads well in the CA/CMC matrix. Liu et al. (2021) found that in a pork freshness film made of polyvinyl alcohol, carboxymethylcellulose sodium, and red cabbage anthocyanin, the crystallinity of the film changes based on the amount of anthocyanin [34].

### 3.4. Microstructure

The SEM images of the surface and cross-section of the CA/CMC complex films and the CA/CMC/PCA indicator films are shown in Figure 4. From the scanning electron microscope images (Figure 4A–E,a–e), it can be seen that due to the good compatibility and film-forming properties of carrageenan and CMC, the cross-sections of the CA/CMC complex films were relatively uniform and smooth. When the amount of PCA increased from 2.5 wt% to 10 wt%, the film surface gradually showed agglomeration phenomenon. As reported by Liu et al. (2021) an increase in the presence of certain components can lead to a looser and rougher structure [35]. In the case of the colorimetric films for red cabbage anthocyanin (RCA) freshness monitoring, Wang et al. (2024) demonstrated that a low concentration of RCAs improved the compatibility of the film matrix, while a higher concentration affected the uniformity of the film [19]. Similar phenomena were observed in the smart biodegradable film containing emblica anthocyanin for fish fillet freshness monitoring by Gasti et al. (2021), where a higher concentration of emblica anthocyanin led to a rougher surface of the complex film [36]. In our study, as the percentage of PCA in the CA/CMC/PCA films increased from 2.5% to 10%, noticeable changes in the surface and cross-section morphology were observed. At lower PCA concentrations (2.5% and 5%), the films showed relatively more uniform surface and cross-section structures, indicating better compatibility between the components. However, as the PCA concentration increased to 7.5% and 10%, the surface became rougher, and the cross-section structure showed signs of non-uniformity, which may be due to the aggregation or interaction of PCA molecules at higher concentrations. These morphological changes may have implications for the performance of the films, such as their barrier properties, mechanical strength, and sensing ability for freshness monitoring.

### 3.5. Optical Characteristics

Color and transparency of film are important indices in food packaging applications. The initial color of the indicator film is determined by the color of the matrix and the indicator [27]. The a* value represents the redness of the film, the b* value represents the yellowness of the film, and the L* value represents the lightness of the film. The color parameters (L*, a*, and b*) and opacity of the films are shown in Table 1. The CA/CMC complex films were transparent and colorless, while the CA/CMC/PCA films presented a visually distinguishable blue-purple color. As the content of PCA increased, the color of the film changed, manifested as an increase in the a* value and a decrease in the b* value. Correspondingly, the indicator film showed a color change trend from light to dark. The increase in anthocyanin content leads to a decrease in the L* value of the CA/CMC/PCA indicator film and an increase in opacity, indicating that the anthocyanin content affects the color appearance of the film [23]. The increase in opacity is due to the dense polymer composite structure formed between carrageenan, CMC, and PCA, which hinders the passage of light through the film, thus increasing the opacity of the film [19]. Qin et al. (2019) also observed that the red color of the cassava starch film based on wolfberry anthocyanins increased with the increase in anthocyanin content, and the a* value increased [37]. Jiang et al. (2020) found that the crystal structure of the complex film based on CMC/starch and purple sweet potato anthocyanins was changed by the electrostatic interaction between starch, CMC, and PSPA, thereby reducing the light transmittance [17].

### 3.6. Thickness and Mechanical Properties

Film thickness is a key characteristic parameter for evaluating the mechanical and light barrier properties of packaging materials [38]. When a composite film is subjected to external forces like stretching, compression, or bending, a thicker material can better disperse and resist these forces. This reduces the possibility of deformation and fracture [26]. From Table 1, it can be seen that as the PCA addition amount increases, the thickness of the composite film gradually increases. Compared with the CA/CMC complex films, when the PCA addition amount is 10%, the thickness of the indicator film increases by 20% (*p* < 0.05). This may be because the addition of PCA leads to an increase in the dry matter content of the indicator film solution, thus causing an increase in the film thickness. Similarly, Crizel et al. (2017) also found that the thickness of the complex film prepared by adding olive powder extract to chitosan increased [25].

Furthermore, as shown in Figure 5, when the PCA concentration in the complex film rises, the film’s tensile strength (TS) drops while its elongation at break (EB) increases. The CA/CMC complex film had the highest TS (11.70 MPa) and the lowest EB (15.34%). The TS values of the CA/CMC/PCA indicator films were close to the purple tomato anthocyanins/chitosan indicator films [39], but significantly lower than the starch/carboxylcellulose/ purple sweet potato anthocyanin indicator films [17]. Compared with the CA/CMC complex film, when the PCA content increases by 10 wt%, the TS value decreases by 18.38%, while the EB value increases by 65.91%. The mechanical property changes occurred because adding PCA to the CA/CMC complex film matrix can function as a plasticizer. Since PCA has good hydrophilicity, it reduces the interaction between macromolecules, causing the TS of the CA/CMC complex film matrix to decrease. Meanwhile, the hydrogen bonds formed between the hydroxyl groups in anthocyanin polyphenols and those in CA/CMC also contribute to the increase in the TS and EB of the complex film network. Gasti et al. (2021) also discovered that adding purple tomato anthocyanin to the chitosan/polyvinyl alcohol complex film or incorporating purple sweet potato anthocyanin glycosides into the chitosan film both result in an increase in the TS and a decrease in the EB of the complex film [36]. This is due to the new hydrogen bonds formed between anthocyanin and polysaccharide.

### 3.7. Water Vapor Permeability (WVP)

WVP is an important indicator for evaluating the water-barrier properties of food packaging films [20]. Generally, good moisture retention benefits food preservation. The polymer structure and the interaction between additives and the polymer are key factors affecting the film’s water permeability. One main function of food packaging materials is to isolate food from surrounding air vapor and prevent or delay food deterioration. So, the film’s WVP should stay low [31]. Figure 6 shows how the PCA addition amount affects the complex film’s WVP.

As the concentration of PCA increased from 0% to 10%, the WVP of the films exhibited an initial decrease followed by an increase. The WVP reached its minimum value of 0.66 g·mm/(m^2^·day·kPa) at a PCA concentration of 2.5%, which was significantly lower (*p* < 0.05) compared to the film without anthocyanins (WVP = 1.13 g·mm/(m^2^·day·kPa)). This indicates that an appropriate amount of PCA can enhance the water resistance of CA/CMC films. The large aromatic rings and pyran rings in the anthocyanin backbone can disrupt the internal network structure of carrageenan-MPE films, thereby reducing the films’ affinity for WVP [18]. Research by Tavares et al. (2019) demonstrated that blending carboxymethyl cellulose (CMC) with starch can reduce the WVP of corn starch films, suggesting that the combination of CMC and starch can improve the water resistance of films to some extent [40]. The findings of this study are consistent with these results, indicating that the composite of PCA with CA/CMC may influence the film’s microstructure and water vapor permeability through a similar mechanism. However, as the anthocyanin concentration further increased to 5%, 7.5%, and 10%, the WVP values gradually increase. This phenomenon may be attributed to the excessive incorporation of anthocyanin molecules, which disrupted the homogeneity of the film. This disruption likely led to the formation of more pores or defects within the film matrix, thereby creating additional pathways for water vapor transmission.

### 3.8. TPC and Antioxidant Activity

Plant polyphenols are excellent antioxidants and play a crucial role in active food packaging. Adding polyphenolic substances to food outer packaging materials can inhibit the oxidation of proteins and lipids, thus extending the shelf life of food [20]. The total phenolic content (TPC) of films with different PCA addition amounts is shown in Figure 7. Compared with the control group, the total phenolic content of the indicator film with added PCA increased significantly. The CA/CMC/PCA composite film with 10% PCA added had the highest TPC. Jiang et al. (2020) found that when apple peel polyphenols were added to a chitosan base to prepare an active food packaging film, the greater the amount of apple peel polyphenols added, the higher the total phenol content and the stronger the antioxidant capacity [17].

Anthocyanins contain a large number of phenolic hydroxyl structures that can provide hydrogen atoms to scavenge free radicals [41]. Free radicals are scavenged by generating phenoxy groups through phenolic hydroxyls, which improves the antioxidant activity of the film [42]. The radical scavenging activity of films with different PCA addition amounts is shown in Figure 7. The CA/CMC films have relatively weak scavenging activity for DPPH and ABTS radicals. When the PCA content in the composite films was 10%, the scavenging rates of DPPH free radicals and ABTS free radicals were 43.27% and 57.52%, respectively. This shows that the anthocyanin content has a significant impact on the antioxidant capacity of the composite film and indicates that the free radical scavenging capacity of the film is proportional to the PCA addition amount. After adding PCA, the radical scavenging activity of the CA/CMC/PCA indicator film increased significantly. This may be because anthocyanin is a phenolic compound capable of supplying H+ ions and facilitating the transformation of free electrons, effectively enhancing the antioxidant capacity [31]. A similar increase in antioxidant activity was also observed when mulberry polyphenol extract rich in anthocyanins was added to carrageenan [18]. Additionally, incorporating purple sweet potato anthocyanins into the chitosan film also had the effect of increasing its antioxidant activity [17].

### 3.9. pH Sensitivity

The CA/CMC/PCA indicator film shows obvious color changes in the buffer solution with pH values ranging from 2.0 to 11.0 (Figure 8). The CA/CMC/PCA indicator films are pink (pH 2.0–3.0), lavender (pH 4.0–6.0), blue (pH 7.0–8.0), and green (pH 9.0–11.0). The pH sensitivity of the CA/CMC/PCA indicator films is mainly caused by the structural changes of anthocyanins in the PCA [33]. The structural changes of anthocyanins at different pH values lead to the color change of the film [17]. The structure of anthocyanins is sensitive to pH changes, thus causing its color to change. In acidic solutions, the structure of anthocyanins is a flavylium cation. It transforms into a quinone base in weak alkaline solutions and becomes structureless in high-pH solutions [17]. Many food spoilage processes are closely related to the pH value changes of food. The microbial spoilage of proteins in most animal products can lead to product deterioration, and the resulting pH range is approximately 6.0–8.0 [33]. From this, we can know that the CA/CMC/PCA indicator film has the potential to become a pH-indicator film for monitoring the freshness of food.

### 3.10. Color Stability

To accurately convey the freshness information of the monitored meat, it is necessary to ensure the color stability of indicator films within food’s shelf life. Figure 9 shows the color stability of indicator films with different formulations (CA/CMC/PCA-2.5%, CA/CMC/PCA-5%, CA/CMC/PCA-7.5%, CA/CMC/PCA-10%) stored under the conditions of 25 °C/light, 25 °C/dark, 4 °C/light, and 4 °C/dark for 10 days.

As can be seen from the figure, under the condition of 25 °C/light, with the increase in storage time, the ΔE values of all films show an upward trend, and the films with higher PCA contents have relatively larger increases in ΔE values. This indicates that high temperature and light accelerate the color change of the films, and high-concentration PCA may make the films more sensitive to environmental factors [43]. Under the condition of 25 °C/dark, the color change of the films is relatively gentle, but it can still be observed that the ΔE values increase to a certain extent over time, indicating that even without light, temperature also affects the color stability of the films. Under the conditions of 4 °C/light and 4 °C/dark, the increase in the ΔE values of the films is significantly smaller than that under the 25 °C condition, which shows that low temperature can significantly slow down the color change of the films. At the same time, by comparing the curves under the conditions of 4 °C/light and 4 °C/dark, it can be found that the influence of light on the color stability of the films still exists at low temperatures, but its influence degree is lower compared with the high-temperature situation. The color stability of the films is jointly affected by temperature, light, and PCA content [44]. Temperature and light are important environmental factors causing the color change of the films, and the increase in PCA content may exacerbate the color change of the films under the action of environmental factors.

### 3.11. CA/CMC/PCA Indicator Film Monitor the Hairtail Freshness

The detection of the change in the freshness of aquatic products during storage and transportation is an important application of pH-sensitive films [30]. The decomposition of protein in the fish meat causes the volatilization of amines and ammonia produced by corrupting bacteria, and the production of amines causes a change in the pH of deteriorated fish [19]. Consequently, in many instances, a pH value exceeding 8.0 is regarded as a robust indicator of fish spoilage [19]. pH is often used in conjunction with other indices, such as the content of total volatile basic nitrogen (TVB-N), trimethylamine (TMA) levels, and sensory evaluations, to provide a comprehensive assessment of fish freshness. Thus, the relationship between the color variation of the smart indicator film and the variation of TVB-N content and pH were analyzed. According to the “Safety Standard for Chinese Seawater Shrimp” (GB 2733-2015), the TVB-N (total volatile basic nitrogen) content of seawater shrimp before consumption should be kept under 30 mg/100 g [19]. CA/CMC/PCA indicator films with different PCA concentrations were placed in petri dishes to indicate the freshness of hairtail samples stored at 4 °C for 0, 3, 5, 7, 9, and 11 days. The higher the PCA concentration in the indicator film, the more pronounced the color change of the indicator film during the refrigeration of the hairtail. Indicator films with low PCA concentrations (2.5%, 5.0%, and 7.5%) exhibit color changes that are difficult to distinguish with the naked eye, whereas indicator films with a high PCA concentration (10%) show significant color changes. Table 2 shows the color changes of the CA/CMC/PCA film with a PCA concentration of 10% during the refrigeration of the hairtail. The color responses of the CA/CMC/PCA indicator films during the storage of the hairtail are exhibited in Table 2. On the 0th day, the initial pH and TVB-N content for the fresh hairtail were 6.38 and 21.98 mg/100 g, respectively (Figure 10), while the color of the CA/CMC/PCA films was the initial blue-purple. On the 7th day, the pH of the hairtail was 7.69, while the TVB-N value was 47.18 mg/100 g, both of which increased to an unacceptable threshold, representing the spoilage of the hairtail. At this time, the color of the CA/CMC/PCA indicator films turned to blue-green. The color response of the CA/CMC/PCA indicator film on the hairtail has good consistency with the pH response of the indicator film in the buffer solution, and when the pH was around 8.0, the color of the films changed to blue-green. Similarly, when using an agar/purple sweet potato anthocyanin indicator film to detect the freshness of pork, it was found that as the pH of pork increased from 5.8 to 7.5, the color of the indicator film shifted from pink to green, indicating severe spoilage of the pork [26]. Additionally, Li et al. (2021) discovered that when employing a purple sweet potato/chitosan indicator film to assess fish freshness, the film’s color transitioned from dark green to light green as the TVB-N value in the fish flesh from 16.7 mg/100 g to 38.8 mg/100 g [39]. Some studies have found that when the indicator film contains an appropriate amount of anthocyanin, the color change is visible to the naked eye. When the anthocyanin content is low, the color change is not obvious. However, an excessive amount of anthocyanin will lead to an increase in the reaction time, resulting in a slow color change of the film [17].

In the test on the hairtail, a distinguishable color change was observed with the test film containing 7.5 wt% PCA, which means that the film at this concentration was effective in real-time monitoring of freshness. In previous studies, the use of anthocyanin in the preparation of indicator films to monitor the freshness of aquatic products has also been reported, such as by Zhang et al. (2019), who prepared a starch/polyvinyl alcohol pH indicator label based on purple sweet potato anthocyanin and red cabbage anthocyanin as indicators for detecting the freshness of shrimp [43]. Krpa and Milenkovic (2019) found that the films prepared by blending bacterial *nanocellulose* and black carrot anthocyanin extract were significantly effective in monitoring the spoilage process of fish [45]. Therefore, CA/CMC/PCA indicator films can also be used to monitor the freshness of perishable foods such as meat and other types of seafood.

## 4. Conclusions

This study represents a significant breakthrough in developing smart bio-based packaging films by incorporating purple cabbage anthocyanin (PCA) into CA/CMC composite films. The abundant availability and low cost of raw materials such as purple cabbage, CA, and CMC, combined with a simple preparation method, make this composite film a promising candidate for large-scale production, offering substantial commercial potential. The films exhibited excellent real-time pH responsiveness and significantly enhanced antioxidant and barrier properties. Notably, the CA/CMC/PCA-10% film effectively monitored hairtail freshness, with distinct color changes, achieving the integration of real-time pH response and superior performance. While this study presents promising results, further optimization of the mechanical properties is needed to enhance the films’ suitability for diverse practical applications. Additionally, food safety risk analysis is an important area for future research. These findings not only advance the development of intelligent packaging but also provide direction for future studies focused on enhancing the performance and scalability of pH-responsive bio-based films for food safety and freshness monitoring.

## Figures and Tables

**Figure 1 foods-14-00694-f001:**
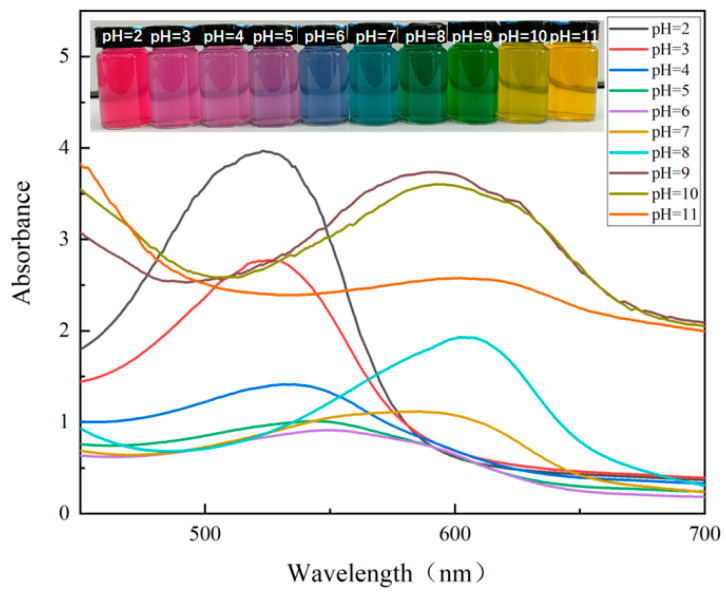
Photographs and visible spectra of PCA solutions prepared at different pH values (2.0–11.0). Note: PCA solution concentration: 1 mg/mL.

**Figure 2 foods-14-00694-f002:**
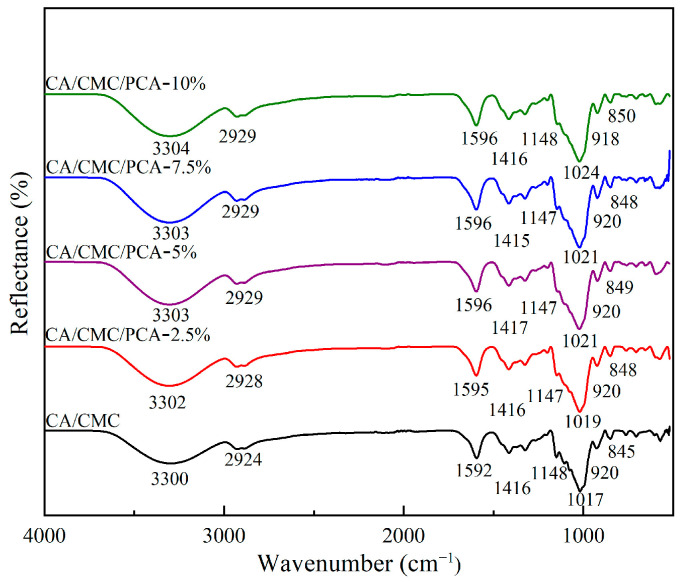
Fourier transform infrared (FTIR) spectra of the CA/CMC, CA/CMC/PCA-2.5%, CA/CMC/PCA-5%, CA/CMC/PCA-7.5%, and CA/CMC/PCA-10% films.

**Figure 3 foods-14-00694-f003:**
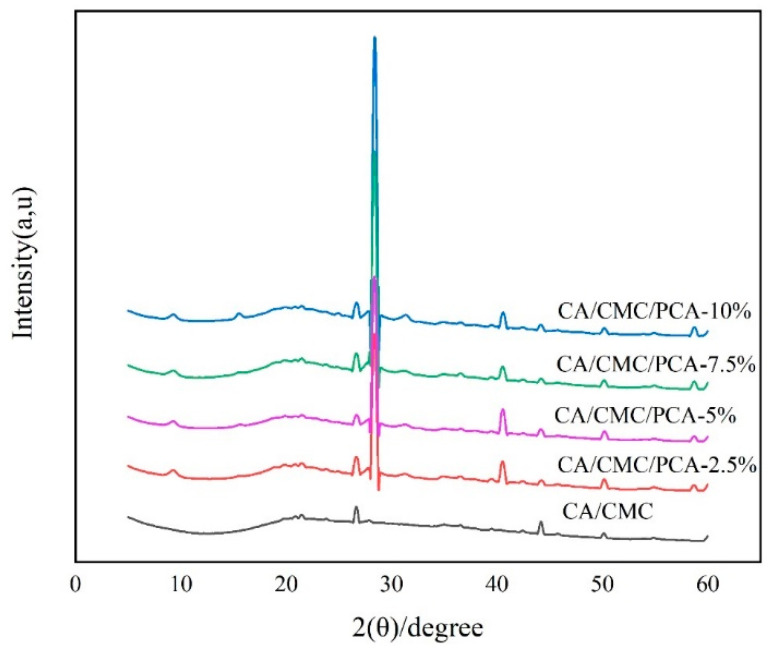
XRD spectra of the CA/CMC, CA/CMC/PCA-2.5%, CA/CMC/PCA-5%, CA/CMC/PCA-7.5%, and CA/CMC/PCA-10% films.

**Figure 4 foods-14-00694-f004:**
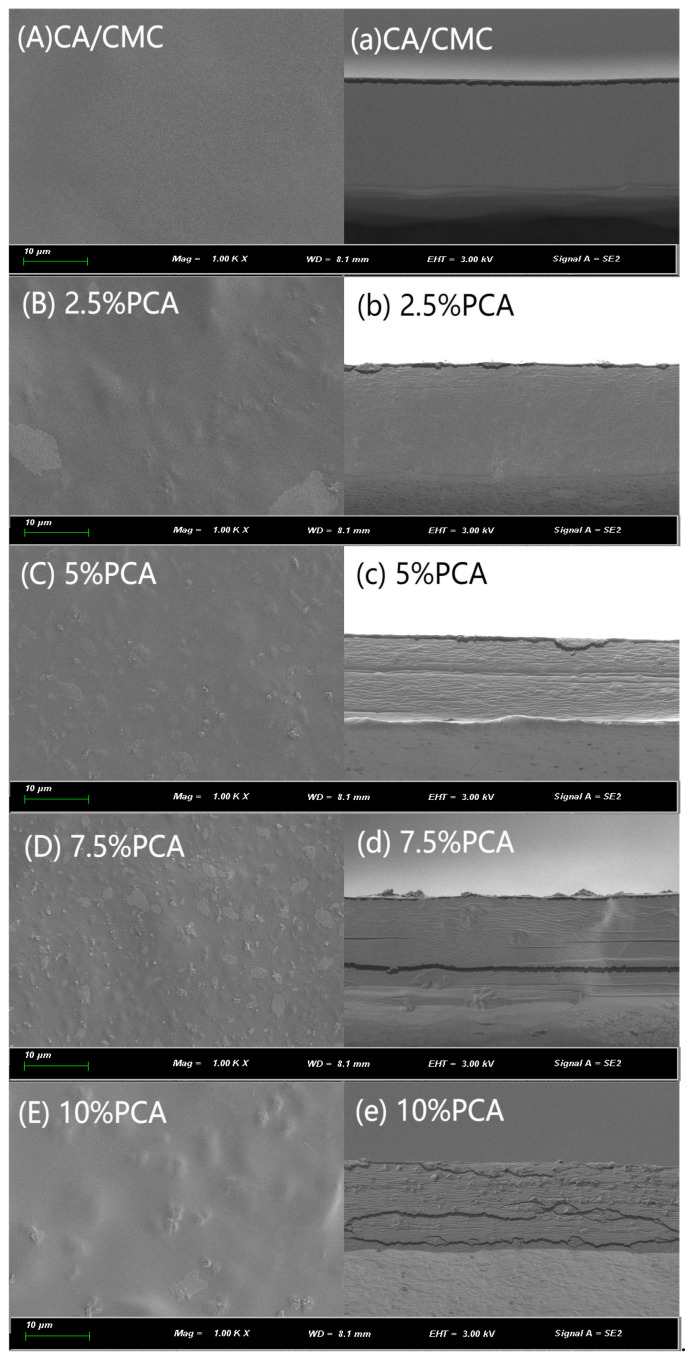
Scanning electron microscopy (SEM) images of cross-section images and surface images of CA/CMC, CA/CMC/PCA-2.5%, CA/CMC/PCA-5%, CA/CMC/PCA-7.5%, and CA/CMC/PCA-10% films. Note: (**A**–**E**) are surface images, (**a**–**e**) are cross-section images.

**Figure 5 foods-14-00694-f005:**
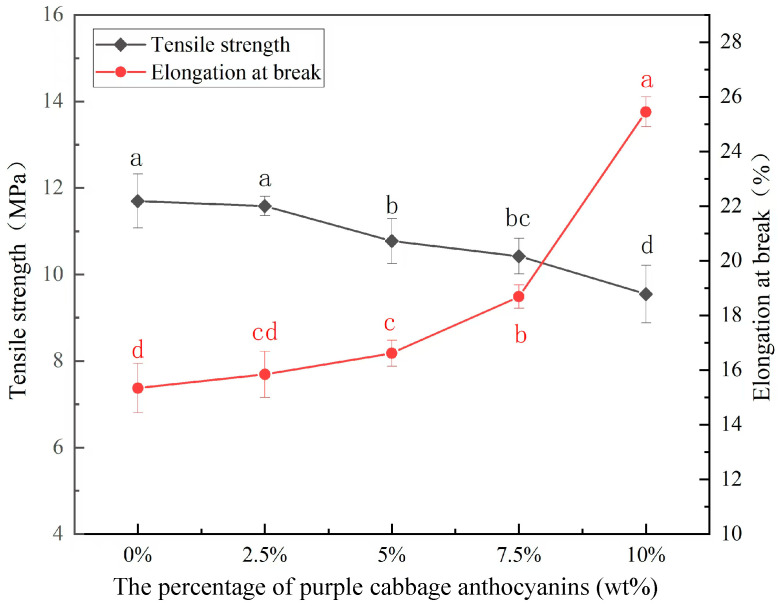
TS and EB of the CA/CMC, CA/CMC/PCA-2.5%, CA/CMC/PCA-5%, CA/CMC/PCA-7.5%, and CA/CMC/PCA-10% films. Note: different letters indicate significant differences between data (*p* < 0.05).

**Figure 6 foods-14-00694-f006:**
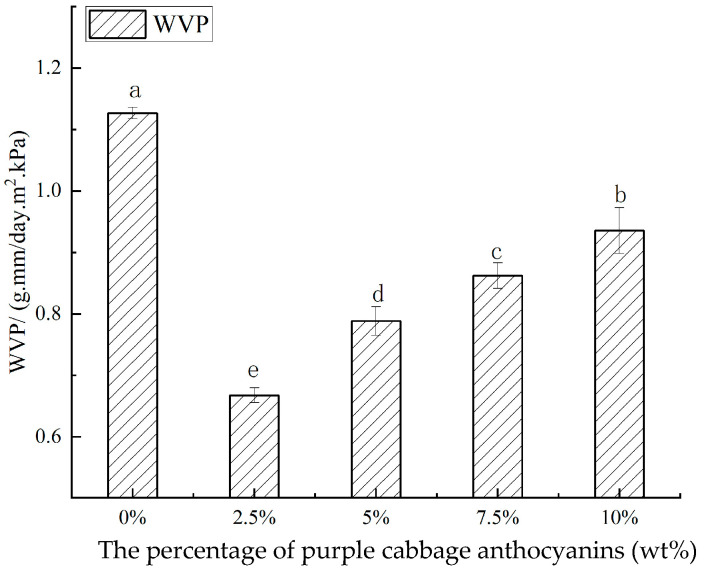
WVP of the CA/CMC, CA/CMC/PCA-2.5%, CA/CMC/PCA-5%, CA/CMC/PCA-7.5%, and CA/CMC/PCA-10% films. Note: different letters indicate significant differences between data (*p* < 0.05).

**Figure 7 foods-14-00694-f007:**
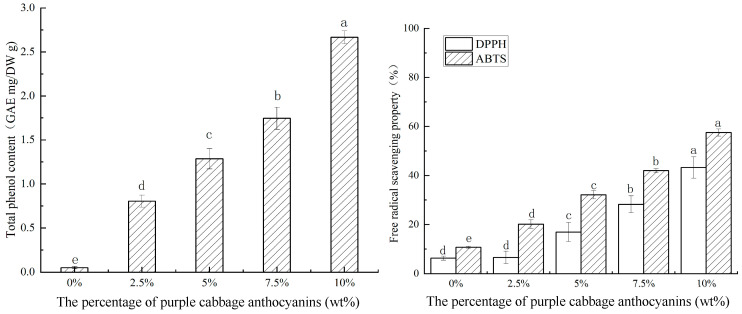
Total phenol content and free radical scavenging activity of the CA/CMC, CA/CMC/PCA-2.5%, CA/CMC/PCA-5%, CA/CMC/PCA-7.5%, and CA/CMC/PCA-10% films. Note: different letters indicate significant differences between data (*p* < 0.05).

**Figure 8 foods-14-00694-f008:**
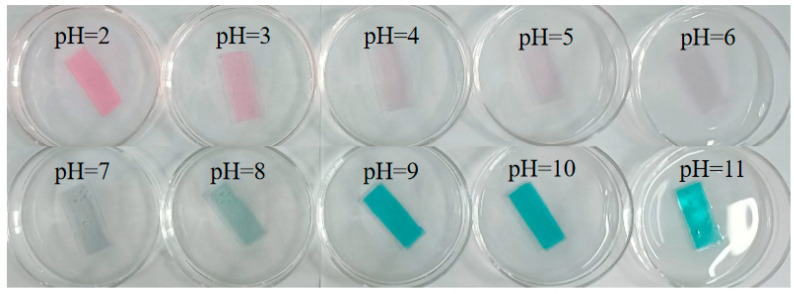
Picture indicator films in different buffer solutions (pH 2~11). Note: The PCA concentration of the indicator film was 10%.

**Figure 9 foods-14-00694-f009:**
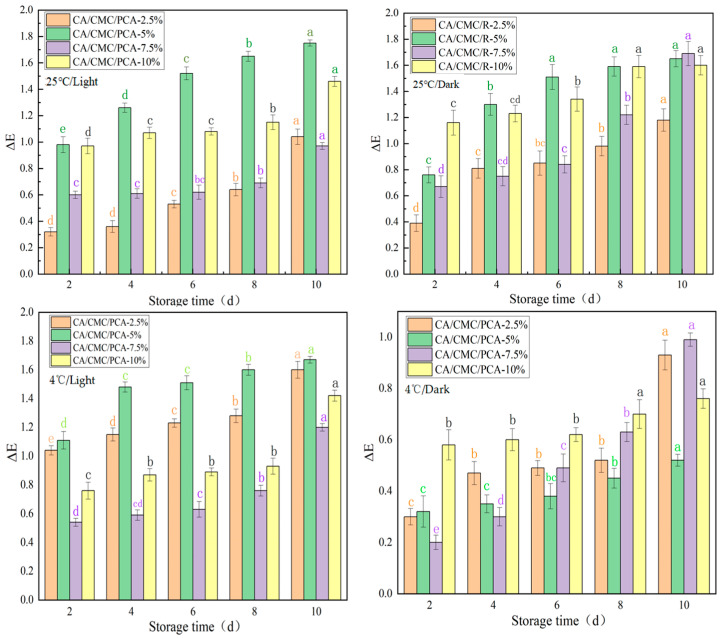
The indicator films are stored at 25 °C/light, 25 °C /dark, 4/℃light, 4 °C/dark for 10 days of color stability. Note: different letters indicate significant differences between data (*p* < 0.05).

**Figure 10 foods-14-00694-f010:**
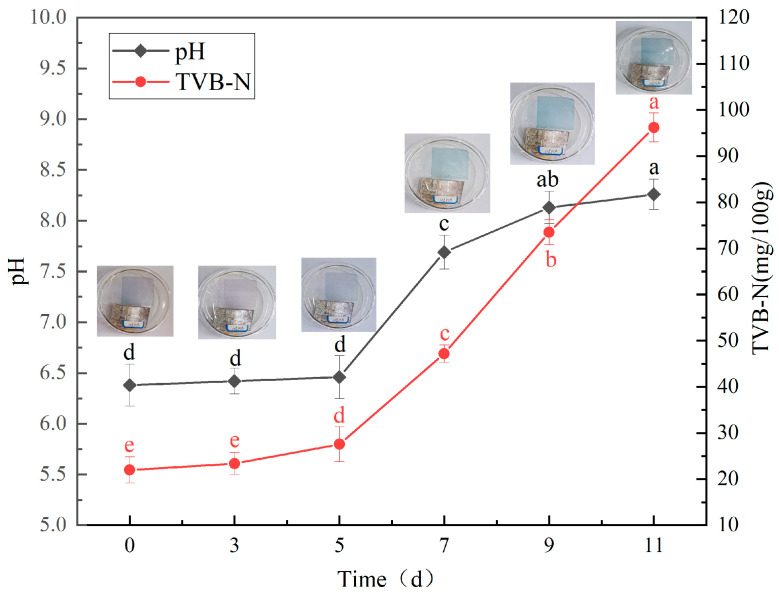
pH and total volatile basic nitrogen (TVB-N) content and pH change during hairtail spoilage. Note: The PCA concentration of the indicator film was 10%. Note: different letters indicate significant differences between data (*p* < 0.05).

**Table 1 foods-14-00694-t001:** Optical properties of CA/CMC, CA/CMC/PCA-2.5%, CA/CMC/PCA-5%, CA/CMC/PCA-7.5%, CA/CMC/PCA-10% films.

Anthocyanin Concentration	Thickness/mm	L*	a*	b*	Opacity/%	Picture
0%	0.050 ± 0.01 ^e^	85.73 ± 0.19 ^a^	0.23 ± 0.05 ^e^	2.03 ± 0.15 ^a^	3.12 ± 0.01 ^e^	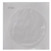
2.5%	0.054 ± 0.01 ^d^	82.10 ± 0.54 ^b^	−2.43 ± 0.09 ^d^	1.67 ± 0.12 ^b^	3.21 ± 0.05 ^d^	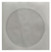
5%	0.057 ± 0.02 ^c^	77.77 ± 0.21 ^c^	0.20 ± 0.08 ^c^	−0.47 ± 0.07 ^c^	3.32 ± 0.02 ^c^	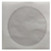
7.5%	0.059 ± 0.02 ^b^	73.57 ± 0.39 ^d^	1.90 ± 0.81 ^b^	−1.63 ± 0.04 ^d^	3.56 ± 0.08 ^b^	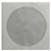
10%	0.060 ± 0.03 ^a^	72.43 ± 0.38 ^e^	4.07 ± 0.09 ^a^	−2.27 ± 0.15 ^e^	4.11 ± 0.04 ^a^	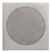

Note: Data with different letters in the same column mean significant difference (*p* < 0.05).

**Table 2 foods-14-00694-t002:** The color response of the CA/CMC/PCA films when monitoring the freshness of the hairtail (stored at 4 °C for 11 days).

	0 d	3 d	5 d	7 d	9 d	11 d
CA/CMC/PCA-2.5%	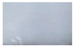	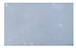	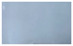	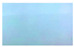	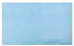	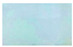
CA/CMC/PCA-5%	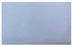	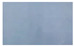	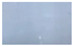	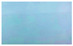	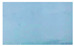	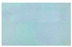
CA/CMC/PCA-7.5%	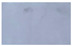	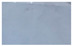	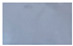	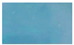	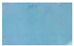	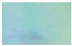
CA/CMC/PCA-10%	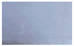	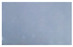	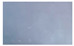	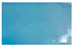	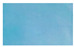	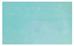

## Data Availability

The original contributions presented in the study are included in the article, further inquiries can be directed to the corresponding author.

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
