# Peer review of "Preparation and Characterization of Antioxidative and pH-Sensitive Films Based on κ-Carrageenan/Carboxymethyl Cellulose Blended with Purple Cabbage Anthocyanin for Monitoring Hairtail Freshness"

_foods, 2025, doi:10.3390/foods14040694_

Round 1
Reviewer 1 Report
Comments and Suggestions for Authors
Dear authors, below are some general considerations with suggestions for improving the work and then some suggestions separated by sections.
The wording duplication percentage in the manuscript is to high, please reduce it.
Please, discuss the advantages and disadvantages of the PCA-based film relative to existing solutions (e.g. curcumin, betalains, or alizarin-based films). How about scalability and commercial viability? The practical application of these films in industrial settings is possible? Manufacturing feasibility, cost, storage conditions, and regulatory compliance should be discussed.
Authors did not consider performing microbial assessments (e.g., bacterial load analysis)? A microbiological validation would provide stronger support for the film’s reliability.
Similarly, how about the long-term stability of the indicator to maintaining color stability for 10 days? no statistical tests (e.g., ANOVA) were conducted to confirm significant differences over time? As well, analyzing degradation rates under different environmental conditions (light exposure, temperature variations) is necessary.
Abstract: instead of listing experimental parameters, authors could also focus on the key findings and implications.
The novelty statement should be clearer: How is this work different from previous studies?
Introduction:
The introduction repeats some background information on pH-sensitive films and polysaccharides multiple times. Streamlining the literature discussion will improve readability.
Some citations are overused without adding new context.
The rationale for selecting PCA should be more clearly explained. What advantages does PCA have over other anthocyanins (e.g., black rice, blueberries)?
Line 62-63: could be rewritten
The hypothesis should be clearly stated at the end of the introduction
MM:
How about the control group indicator film? the study evaluates different PCA concentrations (2.5–10%) but does not include a control group with only κ-carrageenan/CMC (without PCA). A non-PCA film should be tested to isolate the effects of anthocyanin addition.
Sample size: the study only conducts three replicates per experiment. This is too low, particularly for mechanical properties and color stability.
How was the film attached? Did it maintain consistent contact with the headspace?
Sensory evaluation was performed? since the film is intended for consumer use
The materials and methods section is too concise, please detail the methods utilized
Results and discussion: please expand discussion to multiple food matrices to improve generalizability (also in conclusion).
The film's response was tested at 4°C, but how about under different storage conditions (e.g., frozen, ambient, or fluctuating temperatures)? Please discuss this
Could non-pH-related reactions affect film color?
The study did not analyze how humidity affects film strength? despite the well-known moisture sensitivity of polysaccharide-based films?
Table and figure legends need more detail. For example, Figure 8 should clearly explain why certain pH ranges correspond to spoilage.
Some sentences are overly long and complex. Breaking them into shorter, more direct statements would enhance readability.
The results are well-structured, but more quantitative comparisons with previous studies are needed. How does the PCA film perform relative to other pH-sensitive films in terms of sensitivity and response time?
The claim that the film maintains color stability for 10 days is not statistically validated.
Some numerical values in tables do not match the text. Please, double-check data consistency.
Figure 10 should include a statistical significance test (e.g., error bars, p-values) to validate the observed trends with legend information
Ensure that units of measurement are consistent throughout the manuscript (e.g., pH values, TVB-N concentrations).
Conclusion: this section is not making explicit the practical implications of the findings and future research directions. In addition, the conclusion lacks discussion of limitations. Addressing the challenges in industrial application, cost-effectiveness, and potential could improve this section.
Discuss the limitations of the study, including potential challenges in commercial application.
Include a brief discussion on regulatory considerations, as food packaging materials must meet safety standards.
Author Response
Response to reviewer
We sincerely thank reviewer for their valuable feedback that we have used to improve the quality of our manuscript. The reviewer comments are laid out below in boldface font and specific concerns have been numbered. We answered questions pointed out by the reviewers and suggestions made point by point in blue font, and changes / additions to the manuscript are given in red text.
- The wording duplication percentage in the manuscript is to high, please reduce it.
Answer: We have revised the words in the manuscript.
- Please, discuss the advantages and disadvantages of the PCA-based film relative to existing solutions (e.g. curcumin, betalains, or alizarin-based films). How about scalability and commercial viability? The practical application of these films in industrial settings is possible? Manufacturing feasibility, cost, storage conditions, and regulatory compliance should be discussed.
Answer: We have added the relevant information in the introduction section.
- Authors did not consider performing microbial assessments (e.g., bacterial load analysis)? A microbiological validation would provide stronger support for the film’s reliability.
Answer: The focus of our research is on pH-sensitive indicator films. We primarily demonstrate the color changes of the indicator films during the spoilage of hairtail fish. Therefore, the antimicrobial properties of the indicator films were not considered. In future studies, we will analyze both the antimicrobial effects of the indicator films as packaging materials and their color response.
- Similarly, how about the long-term stability of the indicator to maintaining color stability for 10 days? no statistical tests (e.g., ANOVA) were conducted to confirm significant differences over time? As well, analyzing degradation rates under different environmental conditions (light exposure, temperature variations) is necessary.
Answer: We added relevant information in this section and modified Figure 9.
- Abstract: instead of listing experimental parameters, authors could also focus on the key findings and implications.
The novelty statement should be clearer: How is this work different from previous studies?
Answer: Following the recommendations, the abstract section was re-written, highlighting the findings and significance.
Introduction:
- The introduction repeats some background information on pH-sensitive films and polysaccharides multiple times. Streamlining the literature discussion will improve readability.
Some citations are overused without adding new context.
The rationale for selecting PCA should be more clearly explained. What advantages does PCA have over other anthocyanins (e.g., black rice, blueberries)?
Line 62-63: could be rewritten
The hypothesis should be clearly stated at the end of the introduction
Answer: Thank you very much for your comments, we rewritten the introduction, streamlined the references, and provided a brief literature review of natural pH indicators, polysaccharide packaging films, etc.
MM:
- How about the control group indicator film? the study evaluates different PCA concentrations (2.5–10%) but does not include a control group with only κ-carrageenan/CMC (without PCA). A non-PCA film should be tested to isolate the effects of anthocyanin addition.
Answer:In our study, all sample groups were subjected to relevant experimental analyses and compared with the control group. In the hairtail refrigeration experiment, we also conducted experimental analyses on all samples. However, in terms of color response, the color changes in the control and low-concentration groups were not significant. Therefore, only the color photographs of the films with a 10% PCA addition are presented in the manuscript.
- Sample size: the study only conducts three replicates per experiment. This is too low, particularly for mechanical properties and color stability.
Answer: In our study, all sample groups were subjected to relevant experimental analyses and compared with the control group. In the hairtail refrigeration experiment, we also conducted experimental analyses on all samples. However, in terms of color response, the color changes in the control and low-concentration groups were not significant. Therefore, only the color photographs of the films with a 10% PCA addition are presented in the manuscript.
- How was the film attached? Did it maintain consistent contact with the headspace?
Sensory evaluation was performed? since the film is intended for consumer use
The materials and methods section is too concise, please detail the methods utilized
Answer: The experimental methods for this section have been detailed in the manuscript and marked in red font.
- Results and discussion:please expand discussion to multiple food matrices to improve generalizability (also in conclusion).
Answer: Therefore, CA/CMC/PCA indicator films can also be used to monitor the freshness of perishable foods such as meat and other types of seafood.
- The film's response was tested at 4°C, but how about under different storage conditions (e.g., frozen, ambient, or fluctuating temperatures)? Please discuss this
Could non-pH-related reactions affect film color?
Answer: In our experiments, we designed tests to evaluate the color stability of the film under conditions of 4°C and 25°C. The experimental results demonstrated that the color of the indicator film remained stable at 4°C. Based on these findings, we can predict that the anthocyanins in the film will also remain stable under freezing conditions.
- The study did not analyze how humidity affects film strength? despite the well-known moisture sensitivity of polysaccharide-based films?
Answer: This study primarily focused on the color changes of the film in response to variations in environmental pH. When the film was used as an indicator label within the packaging of hairtail fish, despite fluctuations in the humidity of the packaging environment, the film was not subjected to external mechanical forces. Consequently, the mechanical properties of the film were not analyzed under these conditions.
- Table and figure legends need more detail. For example, Figure 8 should clearly explain why certain pH ranges correspond to spoilage.
Answer: We added relevant information in this section and modified Figure 8. The specific contents are as follows: The initial pH of fresh fish typically ranges from 6.0 to 6.5, which is slightly acidic to neutral. This pH range is conducive to the growth of spoilage bacteria. As fish begins to spoil, proteolytic bacteria break down proteins into amino acids and further into amines and ammonia. The production of ammonia (NH₃) increases the pH of the fish tissue, rendering it more alkaline. Consequently, in many instances, a pH value exceeding 8.0 is regarded as a robust indicator of fish spoilage. pH is often used in conjunction with other indices, such as the content of total volatile basic nitrogen (TVB-N), trimethylamine (TMA) levels, and sensory evaluations, to provide a comprehensive assessment of fish freshness.
- Some sentences are overly long and complex. Breaking them into shorter, more direct statements would enhance readability.
Answer: We greatly appreciate your professional review of our articles. We have revised the complex sentence patterns included in the manuscript.
- The results are well-structured, but more quantitative comparisons with previous studies are needed. How does the PCA film perform relative to other pH-sensitive films in terms of sensitivity and response time?
- Answer: We added relevant information to the discussion and marked it in red font, specifically as follows:
  Similarly, when using an agar/purple sweet potato anthocyanin indicator film to detect the freshness of pork, it was found that as the pH of pork increased from 5.8 to 7.5, the color of the indicator film shifted from pink to green, indicating severe spoilage of the pork (Choi et al., 2017). Additionally, Li et al. discovered that when employing a purple sweet potato/chitosan indicator film to assess fish freshness, the film's color transitioned from dark green to light green as the TVB-N value in the fish flesh rose from 16.7 mg/100 g to 38.8 mg/100 g.
- The claim that the film maintains color stability for 10 days is not statistically validated.
Some numerical values in tables do not match the text. Please, double-check data consistency.
Figure 10 should include a statistical significance test (e.g., error bars, p-values) to validate the observed trends with legend information. Ensure that units of measurement are consistent throughout the manuscript (e.g., pH values, TVB-N concentrations).
Answer: We added relevant information in this section and modified Figure 10.
Conclusion: this section is not making explicit the practical implications of the findings and future research directions. In addition, the conclusion lacks discussion of limitations. Addressing the challenges in industrial application, cost-effectiveness, and potential could improve this section.
Discuss the limitations of the study, including potential challenges in commercial application.
Include a brief discussion on regulatory considerations, as food packaging materials must meet safety standards.
Answer: The conclusion section was rewritten as recommended marked in red font. Clarify the significance, limitations, and future research methods.

Reviewer 2 Report
Comments and Suggestions for Authors
General comments
This manuscript proposed the preparation and characterization of antioxidative and pH-sensitive films based on carrageenan/carboxymethyl cellulose blended with purple cabbage anthocyanin for monitoring hairtail freshness. The study is of interest to the field of fish preservation and the experimental work is in general performed well providing new information. However, some information about materials and methods is missing, and clarifications are necessary. Specific comments were given hereinafter.
Specific comments
ABSTRACT
Line 22: Add the conditions needed to color difference was is not visible in 10 day.
INTRODUCTION
Use the word indicator instead of indicative.
Write pH instead of ph
Add info about other studies proposing the use of films indicators for the evaluation of fish fresness using cabagge.
MATERIALS AND METHODS
2.2, 2.3 and 2.4 can be written in a brief way. Since the ,mehods are detailed with an exessive extension,For example: line 104: “for the residue after centrifugation and the centrifugation treatment was carried out again” can be removed. Idem line 135: was placed in a cuvette, and placed in the spectrophotometer for scanning determination.
Line 114, Do not begin a sentence with a number, use for example. An aliquot of...
Line 124: which means by release paper?
RESULTS AND DISCUSSIONs
Paragraph from lines 231-236 and from lines 287-292 can be removed since it mention well known info.
Paragraph from lines 303-320 seems out of place when dealing with Microestructure. Revise this issue.
Remove paragraph from lines 452-457 since it was already mentioned at section 3.1. pH response and UV-VIS spectrum of PCA
The authors mention that storage in darkness is beneficial for the color stability of anthocyanins (line 476) But for application the consumer needs to see the indicator, therefore, it can be expose to light. Discuss this point.
Regarding the application to fish, It is necessary to amplify sensitivity of color changes around the TVB-N value of 30. The color shown at day 5 suggest that the film proposed did not possess a siginificant sensititivy to detect spoilage.
Author Response
Response to reviewer
We sincerely thank reviewer for their valuable feedback that we have used to improve the quality of our manuscript. The reviewer comments are laid out below in boldface font and specific concerns have been numbered. We answered questions pointed out by the reviewers and suggestions made point by point in blue font, and changes / additions to the manuscript are given in red text.
ABSTRACT
- Line 22: Add the conditions needed to color difference was is not visible in 10 day.
Answer: The indicative film has good environmental stability, and the color difference is not visible in 10 day in the dark and 4℃ conditions.
INTRODUCTION
- Use the word indicator instead of indicative.
Answer: We have revised the word in the manuscript.
- Write pH instead of ph
Answer: Answer: We feel sorry for our carelessness. We modified this error and marked it in red font.
- Add info about other studies proposing the use of films indicators for the evaluation of fish fresness using cabagge.
Answer:We have added relevant information in the Introduction section.
MATERIALS AND METHODS
- 2, 2.3 and 2.4 can be written in a brief way. Since the ,mehods are detailed with an exessive extension,For example: line 104: “for the residue after centrifugation and the centrifugation treatment was carried out again” can be removed. Idem line 135: was placed in a cuvette, and placed in the spectrophotometer for scanning determination. Line 114, Do not begin a sentence with a number, use for example. An aliquot of...
Answer: We greatly appreciate your professional review of our articles. We have modified and simplified the experimental methods section.
- Line 124: which means by release paper?
Answer: Upon reviewing relevant literature, the term "release paper" is the most commonly used expression. In industrial production, particularly in the fields of adhesive products, composite material manufacturing, and food packaging, "release paper" is widely employed to refer to papers that prevent adhesive materials from sticking together and facilitate subsequent separation operations.
RESULTS AND DISCUSSIONs
- Paragraph from lines 231-236 and from lines 287-292 can be removed since it mention well known info.
Answer: We greatly appreciate your professional review of our articles. We have already deleted this content.
- Paragraph from lines 303-320 seems out of place when dealing with Microestructure. Revise this issue.
Answer: We greatly appreciate your professional review of our articles. We have revised this part of the manuscript and marked it in red font.
- Remove paragraph from lines 452-457 since it was already mentioned at section 3.1. pH response and UV-VIS spectrum of PCA
Answer: We greatly appreciate your professional review of our articles. We have already deleted this content.
- The authors mention that storage in darkness is beneficial for the color stability of anthocyanins (line 476) But for application the consumer needs to see the indicator, therefore, it can be expose to light. Discuss this point.
Answer: We have added the relevant information to this section.The specific contents are as follows: Temperature and light are important environmental factors causing the color change of the films, and the increase in PCA content may exacerbate the color change of the films under the action of environmental factors.
- Regarding the application to fish, It is necessary to amplify sensitivity of color changes around the TVB-N value of 30. The color shown at day 5 suggest that the film proposed did not possess a siginificant sensititivy to detect spoilage.
Answer: We have revised Figure 10 to clearly illustrate the color changes of the indicator film.

Reviewer 3 Report
Comments and Suggestions for Authors
This study synthesized a cast film of CA/CMC/PA as a pH-sensitive film, characterized its properties, and applied it to monitor the freshness of fish. The content is suitable for this journal and is interesting. Please make the following corrections and additions.
1) Line 62: ph → pH
2) Line 74~79: Please define purple cabbage anthocyanin (PCA) here. PCA suddenly appears in Line 82 without a definition.
3) Line 81-83: Why is there no application of the composite film based on CA/CMC? What are the issues involved?
4) Section 3.1 & Fig.1: Please describe the PCA concentration of the sample data shown in Fig. 1 in the caption and main text.
5) Section 3.1 & Fig.1: Up to 3.11, the effect of the PCA concentration on the film's color change is unclear. Shouldn't the VIS spectra of each PCA concentration be shown here before applying it to fish? Please show the relationship between the PCA concentration and the magnitude and sensitivity of the film's color change.
6) Fig.6 and Section 3.7: 'As the PCA concentration in the complex film increases, the WVP value of the complex film decreases.' However, the results shown in the graph show the opposite trend. PCA 10% has the highest value, and 0% is even higher. Please check the description and graph.
7) Section 3.9 & Fig.8: There is no description of the PCA concentration of the samples shown in the graph here either. Please also describe the effect of the PCA concentration.
Author Response
Response to reviewers
We sincerely thank reviewer for their valuable feedback that we have used to improve the quality of our manuscript. The reviewer comments are laid out below in boldface font and specific concerns have been numbered. We answered questions pointed out by the reviewers and suggestions made point by point in blue font, and changes / additions to the manuscript are given in red text.
1.Line 62: ph → pH
Answer: We feel sorry for our carelessness. We modified this error and marked it in red font.
2.Line 74~79: Please define purple cabbage anthocyanin (PCA) here. PCA suddenly appears in Line 82 without a definition.
Answer: We defined PCA here and marked in red font.
3. Line 81-83: Why is there no application of the composite film based on CA/CMC? What are the issues involved?
Answer: Thank you for your valuable comments. In response to your concerns regarding the application of CA/CMC composite films, we have revised the introduction section and further clarified the innovative aspects and objectives of our research. In the original text, we emphasized that although some researchers have developed biodegradable films based on CA or CMC, the application of CA/CMC composite films incorporating purple cabbage anthocyanin (PCA) as a pH-responsive indicator film has not been thoroughly investigated or developed. Therefore, the focus of this study is to introduce PCA into CA/CMC composite films to develop an intelligent bio-based packaging film capable of real-time monitoring of food spoilage. We hypothesize that the incorporation of PCA will significantly enhance the pH sensitivity and antioxidant properties of the composite film, thereby effectively monitoring the spoilage of foods such as hairtail fish. In the revised introduction, we have further elaborated on the fundamental properties of CA and CMC and the potential of their composites to improve the mechanical properties of films. We have also detailed the effects of PCA incorporation on the morphology, structure, optical properties, mechanical performance, antioxidant activity, and pH-responsive color-changing characteristics of the composite film. Additionally, we have discussed the practical application prospects of this composite film in food packaging. Through these revisions, we aim to more clearly convey the innovation of this study and its potential contributions to the field of intelligent packaging. Once again, we appreciate your feedback and believe that these modifications more accurately present the significance and value of our research.
4. Section 3.1 & Fig.1: Please describe the PCA concentration of the sample data shown in Fig. 1 in the caption and main text.
Answer: We added the PCA concentration to the pH sensitivity assay method for PCA and indicated the PCA concentration in Figure 1.
5.Section 3.1 & Fig.1: Up to 3.11, the effect of the PCA concentration on the film's color change is unclear. Shouldn't the VIS spectra of each PCA concentration be shown here before applying it to fish? Please show the relationship between the PCA concentration and the magnitude and sensitivity of the film's color change.
Answer: We have added this part in the manuscript, specifically as follows:
The sensitivity of PCA's color response in solutions of different pH values is closely related to its concentration. In aqueous solutions, a low concentration of PCA (1 mg/mL) can already exhibit significant color changes. Since the color of PCA extract is deep purple, high concentrations of PCA solution can mask the color changes under different pH conditions.
Similarly, the content of PCA in the indicator film is closely related to changes in the environmental pH value. Our research indicates that when CA/CMC/PCA indicator films with different PCA concentrations are placed in petri dishes, the higher the PCA concentration in the indicator film, the more pronounced the color change of the film under different pH conditions. Indicator films with low PCA concentrations (2.5%, 5.0%, and 7.5%) exhibit color changes that are difficult to distinguish with the naked eye, whereas indicator films with a high PCA concentration (10%) show significant color changes.
6.Fig.6 and Section 3.7: 'As the PCA concentration in the complex film increases, the WVP value of the complex film decreases.' However, the results shown in the graph show the opposite trend. PCA 10% has the highest value, and 0% is even higher. Please check the description and graph.
Answer: We have revised this part in the manuscript, specifically as follows: Compared to the CA/CMC complex film, the WVP of the indicator film with added PCA was significantly reduced (p < 0.05). It is worth noting that the WVP of indicator films with different PCA concentrations decreased with the increase of PCA concentration.
7.Section 3.9 & Fig.8: There is no description of the PCA concentration of the samples shown in the graph here either. Please also describe the effect of the PCA concentration.
Answer: We have revised this part in the manuscript, specifically as follows: CA/CMC/PCA indicator films with different PCA concentrations were placed in petri dishes to indicate the freshness of hairtail samples stored at 4°C for 0, 3, 5, 7, 9, and 11 days. The higher the PCA concentration in the indicator film, the more pronounced the color change of the indicator film during the refrigeration of hairtail. Table 2 shows the color changes of the CA/CMC/PCA film with a PCA concentration of 10% during the refrigeration of hairtail.

Round 2
Reviewer 1 Report
Comments and Suggestions for Authors
The authors responded to my suggestions and they have addressed all the comments appropriately. In my opinion, the manuscript is now ready for acceptance.
Author Response
We sincerely thank reviewer for their valuable feedback that we have used to improve the quality of our manuscript.
Reviewer 3 Report
Comments and Suggestions for Authors
Line 381-382: 'Notably, the WVP of indicator films with different PCA concentrations dropped as the PCA concentration increased.'
Is this sentence correct?
Fig. 6 shows that the WVP increases significantly as the PCA concentration increases.
Doesn't this contradict the statement in Lines 388-391, 'However, as the PCA amount increases, the WVP value gradually rises but is still lower than that of the CA/CMC complex film.'?
Author Response
Response to reviewer
We sincerely thank reviewer for their valuable feedback that we have used to improve the quality of our manuscript. The reviewer comments are laid out below in boldface font and specific concerns have been numbered. We answered questions pointed out by the reviewers and suggestions made point by point in blue font, and changes / additions to the manuscript are given in red text.
- Line 381-382: 'Notably, the WVP of indicator films with different PCA concentrations dropped as the PCA concentration increased.
Is this sentence correct?
6 shows that the WVP increases significantly as the PCA concentration increases.
Doesn't this contradict the statement in Lines 388-391, 'However, as the PCA amount increases, the WVP value gradually rises but is still lower than that of the CA/CMC complex film.'?
Answer: We feel sorry for our carelessness. We modified this error and marked it in red font, specifically as follows:
As the concentration of PCA increased from 0% to 10%, the WVP of the films exhibited an initial decrease followed by an increase. The WVP reached its minimum value of 0.66 g·mm/(m²·day·kPa) at a PCA concentration of 2.5%, which was significantly lower (p < 0.05) compared to the film without anthocyanins (WVP=1.13 g·mm/(m²·day·kPa)). This indicates that an appropriate amount of PCA can enhance the water resistance of CA/CMC films. The large aromatic rings and pyran rings in the anthocyanin backbone can disrupt the internal network structure of carrageenan-MPE films, thereby reducing the film's affinity for WVP (Zhang et al., 2020). Research by Tavares et al. (2019) demonstrated that blending carboxymethyl cellulose (CMC) with starch can reduce the WVP of corn starch films, suggesting that the combination of CMC and starch can improve the water resistance of films to some extent. The findings of this study are consistent with these results, indicating that the composite of PCA with CA/CMC may influence the film's microstructure and water vapor permeability through a similar mechanism. However, as the anthocyanin concentration further increased to 5%, 7.5%, and 10%, the WVP values gradually increase. This phenomenon may be attributed to the excessive incorporation of anthocyanin molecules, which disrupted the homogeneity of the film. This disruption likely led to the formation of more pores or defects within the film matrix, thereby creating additional pathways for water vapor transmission.

Round 3
Reviewer 3 Report
Comments and Suggestions for Authors
The revision was well done. This paper is acceptable.